# Effect of insulin treatment on pulsatility ratio and resistance index of the retinal artery in patients with type 2 diabetes

**Tsuneaki Omae** [ID]*, **Youngseok Song, Takafumi Yoshioka, Tomofumi Tani, Akitoshi Yoshida**

Department of Ophthalmology, Asahikawa Medical University, Asahikawa, Japan

* oomae@asahikawa-med.ac.jp

**Data Availability Statement:** All relevant data are within the paper.

**Funding:** The author(s) received no specific funding for this work.

## Abstract

This study aimed to evaluate whether long-term insulin treatment is associated with abnormalities in retinal circulation in type 2 diabetic patients. We evaluated 19 eyes of nondiabetic individuals and 68 eyes of type 2 diabetic patients. The eyes of diabetic patients were classified into two groups according to the presence or absence of long-term insulin therapy. We used a Doppler optical coherence tomography flowmeter to measure diameter, velocity, and blood flow in the major temporal retinal artery. The pulsatility ratio (PR) and resistance index (RI), indices of vascular rigidity, were calculated from the blood velocity profile. PR and RI were significantly elevated in type 2 diabetic patients compared with nondiabetic subjects ($P$ < 0.05). In type 2 diabetes patients, PR and RI were significantly higher in patients receiving long-term insulin treatment than in those without ($P$ < 0.01). There was a significant difference in velocity ($P$ < 0.05), but not diameter and blood flow, between nondiabetic subjects and type 2 diabetes patients. No significant difference in diameter, velocity, or blood flow was observed between the groups with and without long-term insulin treatment. Long-term insulin treatment can affect PR and RI, which might be associated with vascular rigidity of the retinal artery in patients with type 2 diabetes.

## Introduction

Diabetic retinopathy (DR) is a leading cause of blindness, but the mechanisms underlying the development and progression of DR remain unclear. Long-term hyperglycemia can induce an inflammatory response [1], which is involved in the pathogenesis of atherosclerosis aggravation [2] and arterial stiffness [3]. Our previous clinical study on patients with type 2 diabetes showed through laser Doppler velocimetry that the resistive index and pulsatility ratio (PR), which are the indicators of vascular compliance and vascular resistance, of the retinal arterioles in patients with early DR were greater than control subjects [4]. Thus, an abnormality in the vascular rigidity of the retinal arteries may be implicated in the pathogenesis of the development and progression of DR. However, the cause of this altered retinal arterial rigidity in patients with type 2 diabetes with early-stage DR is unclear.

**Competing interests:** The authors have declared that no competing interests exist.

One meta-analysis reported that the use of insulin, one of the medications used to treat diabetes mellitus, is related to the development of DR [5]. Experimental investigation indicated that insulin stimulated the proliferation of vascular smooth muscle cells, which is the prominent hallmark of atherosclerosis [2], in patients with insulin resistance [6]. Moreover, a clinical study demonstrated that insulin-induced decreased vascular compliance under physiological conditions was diminished in patients with type 2 diabetes [7]. Thus, the change in retinal arterial stiffness by insulin treatment may be implicated in the development and progression of DR in patients with type 2 diabetes. Although previous research about relationship between daily dose of insulin and retinal circulation has been investigated in type 1 diabetes patients [8], effect of insulin treatment in retinal circulation including arterial stiffness in type 2 diabetes patients occupied the majority of diabetes is unclear.

Optical coherence tomography (OCT) is a type of noninvasive, rapid, and high-resolution imaging technique based on low-coherence light interferometry and is widely used in the diagnosis and management of ocular disorders. A novel velocimetry strategy using OCT technology with a short acquisition time, namely, Doppler OCT (DOCT), has been developed recently [9]. We have also recently developed a novel type of DOCT velocimetry based on the segmental scanning technique, namely, DOCT flowmetry [10], and the further use of pulse waveform measured with the DOCT flowmeter system enables the evaluation of the rigidity of the retinal arteries [11].

The purpose of this study was to examine retinal arterial parameters including retinal arterial stiffness in patients with type 2 diabetes using this flowmeter. In addition, we categorized patients with type 2 diabetes into two groups according to insulin use and conducted a further comparison among these groups.

## Methods

### Subjects

Overall, 68 consecutive patients (men, 37; women, 31; mean ± SD age, 60.9 ± 9.2 years; range, 41–77 years) with type 2 diabetes mellitus and 19 healthy age-matched control subjects (men, 8; women, 11; mean ± SD age, 57.8 ± 7.9 years; range, 41–71 years) were included. This study complied with the Declaration of Helsinki, and the study protocol was approved by the ethics committee of the Asahikawa Medical University. Each participant received a complete explanation of the study design and protocol and provided written informed consent before enrollment.

Diabetes mellitus was diagnosed in accordance with the following criteria of the American Diabetes Association [12]: fasting blood glucose level >126 mg/dL or the use of oral hypoglycemic agents or insulin. Blood pressure (BP) level exceeding 140/90 mm Hg or the use of antihypertensive medications was considered as hypertension. The condition of patients with plasma total cholesterol level of >220 mg/dL, plasma low-density lipoprotein (LDL) cholesterol level of >130 mg/dL, or both or the use of cholesterol-lowering medications was considered as dyslipidemia. Venous blood was obtained from all participants on the day of the study. Random plasma glucose, hemoglobin A1c (HbA1c), blood urea nitrogen, serum creatinine, and lipid profiles (total, high-density lipoprotein, and LDL cholesterol, as well as triglycerides) were measured immediately using standard methods. Patients with uncontrolled diabetes (HbA1c, >10.0%), poorly controlled hypertension, or history of renal dysfunction or cardiovascular diseases were excluded from the study.

All patients underwent clinical and laboratory assessments and ophthalmologic examinations, including review of their medical history, slit-lamp biomicroscopy, intraocular pressure (IOP) measurement with Goldmann applanation tonometry, and funduscopic examination

using a 90-diopter lens. Mean arterial BP and heart rate were measured using an electronic sphygmomanometer (EP-88Si; Colin, Tokyo, Japan). The ocular perfusion pressure (OPP) was calculated as 2/3 mean arterial BP–IOP. The pupil was dilated with a drop of 0.4% tropicamide (Mydrin M; Santen Pharmaceuticals, Osaka, Japan). The retinopathy levels for each eyes were determined by the maximal grade in any of the seven standard photographic fields and classi-fied as none, mild nonproliferative, moderate-to-severe nonproliferative, or proliferative [13]. We excluded patients with moderate-to-severe nonproliferative DR and proliferative DR. The eye with the worse retinopathy that met inclusion criteria was included. If both eyes had equal retinopathy, one eye was randomly assigned to the study. Patients with other ocular disease, such as glaucoma or retinal vessel occlusion, a history of laser treatment or intraocular surgery, moderate-to-severe cataract, previous ocular inflammation, clinically significant macular edema, and moderate-to-high refractive error (± >3.0 D) were excluded [14, 15].

Patients with type 2 diabetes were categorized into two groups based on their insulin use: those with current insulin treatment (insulin group) or those without insulin treatment (non-insulin group). In addition, the insulin group was defined as those patients receiving insulin therapy for at least 1 year within the duration of diabetes and who were currently on insulin therapy, with a stable regimen for the past 3 months [16, 17]. Previous short-term insulin treat-ment (maximum of 14 days) was permitted.

## DOCT flowmeter

The methodology of this system has been described previously [10, 18]. Briefly, the DOCT flowmeter, which was based on a commercially available spectral-domain OCT system (3D OCT-2000 FA, Topcon Corp., Tokyo, Japan), operated at a wavelength range of 800 nm and was used to perform examinations using the segmental scanning method. The image-captur-ing software was modified for Doppler imaging, and the image-processing software was newly developed to measure retinal circulation. A total of 180 data sets were captured for approxi-mately 2 s with a scan length of 1 mm. Multiple vessels with a diameter >60 μm were detected automatically. Depending on the Doppler angle, the flow rate for measuring within vessels was overestimated. Therefore, measured data with a Doppler angle of <85˚ were included.

The method of measuring retinal circulation using this flowmeter has been described previ-ously [10, 18]. Briefly, a single-color fundus image centered at the optic disc was captured, and then the retinal arteriole at either the superior or inferior temporal site located one disc diame-ter away from the optic disc was chosen manually. The scan location was also set in this proce-dure to ensure it was almost perpendicular to the blood vessel. Flow velocity was calculated with the Doppler angle between the incident beam and flow direction and the amount of Doppler shift (phase image). The Doppler angle was calculated by parallel to each other but perpendicular two scans to the blood vessel with Y = 100 μm apart. The Doppler angle was cal-culated by determining the vessel center of the phase image. The measurement of the cardiac cycle was performed for ~2 s. During this scan, the scan location was adjusted accordingly with the motion-tracking function. During this ~2-s measurement, we were able to observe at least one cardiac cycle pulsation curve. The Doppler shift image (phase image) was calculated by taking the phase difference between adjacent scans as described previously. Collected phase images were registered and summed to find the vessel location, its center, and diameter by the automated software. The flow was then calculated with the vessel diameter and velocity.

To examine the rigidity of the retinal artery, the PR and resistance index (RI), which repre-sent vascular compliance and peripheral resistance, respectively, were calculated from the velocity profile. The maximum velocity ($V_{max}$) and minimum velocity ($V_{min}$) were identified

in the systolic and diastolic phases during a single cardiac cycle, respectively [11]. PR was expressed as $V_{max}/V_{min}$, and RI was expressed as $(V_{max} - V_{min})/V_{max}$.

### Statistical analysis

Comparisons for continuous variables between the groups were performed using unpaired $t$-test with Welch's correction. On the other hand, comparisons for categorical variables were performed using the chi-square test. We analyzed the standardized regression coefficients from the multiple regression analysis of the retinal circulatory parameters in relation to various factors, including insulin treatment. For this analysis, based on our previous studies [4] and previous studies on the retinal pulse wave parameter [19], clinically important variables including traditional DR risk factors were included: age, HbA1c, diabetes duration, body mass index, BP, LDL, estimated glomerular filtration rate, treatment of diabetes (presence or absence of insulin treatment), and vessel diameter. Second, variables with a $P$-value of $<$ .2 from the Pearson analysis were included in the multiple regression analysis. Before enrolling factors for multivariate analysis, we also performed correlation analyses between each pair of all significant factors to exclude multicollinearity. Two factors that showed an absolute value of the correlation coefficient of $>0.7$ were defined as significantly correlated. Among these two factors, we selected the factor with the higher correlation coefficient for the ratio [20].

## Results

The clinical and hemodynamic characteristics of nondiabetic subjects and diabetic patients are shown in Tables 1 and 2. Glucose, HbA1c, and body mass index were significantly higher in the diabetic group than in the control group. Compared with the control group, blood velocity, but not vessel diameter or retinal blood flow, was significantly lower in the diabetic group. Pulse wave analysis revealed a decrease in $V_{min}$, but not in $V_{max}$, in the diabetic group and a greater PR and RI in the diabetic group than that in the control group.

The clinical and biochemical parameters of nondiabetic subjects and diabetic patients as compared among those with or without insulin treatment are presented in Table 3. Retinal vessel parameter as measured by DOCT flowmeter revealed that there were no significant differences in diameter, velocity, or retinal blood flow between the insulin group and noninsulin group (Table 4). However, pulse waveform analysis revealed that PR and RI in the insulin group were significantly increased with elevated $V_{max}$ as compared with the noninsulin group, respectively.

PR and RI were positively related to age, diabetes duration, and insulin use (PR: $P < 0.001$, $P = 0.002$, $P < 0.001$; RI: $P < 0.001$, $P = 0.003$, $P = 0.004$, respectively) and negatively correlated with diastolic BP and mean BP (PR: $P = 0.007$, $P = 0.02$; RI: $P < 0.001$, $P = 0.003$, respectively) by Pearson correlation analysis (Table 5). Multiple regression analysis showed that the PR and RI were positively correlated with age and diabetes duration (PR: $P < 0.001$, $P = 0.03$; RI: $P < 0.001$, $P = 0.04$, respectively), and PR was positively correlated with insulin use in type 2 diabetic patients ($P = 0.002$; Table 6).

## Discussion

Measuring the pulse waveform parameters in the retinal artery is considered a useful evaluation of retinal arterial properties, including vascular resistance and stiffness [11, 21]. In the present study, PR and RI obtained from pulse waveform analysis were significantly greater in patients with type 2 diabetes with early retinopathy than in nondiabetic subjects. In addition, we demonstrated decreased velocity in patients with type 2 diabetes, suggesting increased vascular resistance distal to the retinal artery in patients with type 2 diabetes. The finding of

**Table 1. Characteristics of control subjects and patients with DM.**

| Variable | Control subjects (n = 19) | Patients with DM (n = 68) | P-value* |
|---|---|---|---|
| Mean age, years | 57.8 ± 7.9 | 60.9 ± 9.2 | 0.08 |
| Men/women | 8/11 | 37/31 | 0.33 |
| Glucose, mg/dL | 109.6 ± 27.0 | 146.8 ± 41.1 | <0.001 |
| HbA1c, % | 5.7 ± 0.3 | 6.9 ± 0.8 | <0.001 |
| Duration of diabetes, years | | 9.3 ± 6.0 | |
| BMI, kg/m$^2$ | 22.1 ± 2.4 | 25.2 ± 5.6 | 0.02 |
| Systolic BP, mm Hg | 123.1 ± 9.7 | 131.4 ± 16.0 | 0.08 |
| Diastolic BP, mm Hg | 77.2 ± 7.5 | 76.7 ± 9.8 | 0.26 |
| Mean BP, mm Hg | 92.5 ± 7.6 | 94.9 ± 10.2 | 0.48 |
| Heart rate, beats/min | 71.1 ± 9.7 | 77.5 ± 11.6 | 0.06 |
| IOP, mm Hg | 14.7 ± 2.8 | 15.3 ± 2.8 | 0.24 |
| OPP, mm Hg | 46.9 ± 4.8 | 47.9 ± 7.9 | 0.70 |
| Total cholesterol, mg/dL | 198.5 ± 26.2 | 189.0 ± 26.9 | 0.13 |
| Triglycerides, mg/dL | 112.6 ± 47.0 | 131.1 ± 76.6 | 0.86 |
| HDL, mg/dL | 62.5 ± 12.1 | 57.4 ± 14.2 | 0.11 |
| LDL (mg/dL) | 112.3 ± 21.6 | 107.8 ± 26.2 | 0.26 |
| Blood urea nitrogen, mg/dL | 12.9 ± 3.1 | 15.1 ± 4.2 | 0.06 |
| Creatinine, mg/dL | 0.68 ± 0.14 | 0.72 ± 0.16 | 0.18 |
| eGFR, mL/min/1.73 m$^2$ | 80.0 ± 11.9 | 78.2 ± 14.0 | 0.28 |

Abbreviations: BMI, body mass index; BP, blood pressure; DM, diabetes mellitus; eGFR, estimated glomerular filtration rate; HbA1c, hemoglobin A1c; HDL, high-density lipoprotein; IOP, intraocular pressure; LDL, low-density lipoprotein; OPP, ocular perfusion pressure.

*$P$ values comparing control subjects and patients with diabetes.

elevated vascular resistance in patients with early DR was consistent with our previous observation using laser Doppler velocimetry [4]. Thus, this abnormality of retinal arterial property has already been detected in the early stage of retinopathy. Because our study using laser Doppler velocimetry also indicated that PR was significantly greater, without significant differences in the vessel diameter, blood velocity, and retinal blood flow, in patients with nonproliferative DR compared with those without DR [22], not only elevated vascular resistance but also increased retinal stiffness might be implicated in the development and progression of DR.

**Table 2. Retinal circulatory parameters between control subjects and patients with DM.**

| Variable | Control subjects (n = 19) | Patients with DM (n = 68) | P-value* |
|---|---|---|---|
| Vessel diameter, μm | 100.1 ± 17.1 | 99.8 ± 17.1 | 0.90 |
| Blood velocity, mm/s | 19.0 ± 4.9 | 16.4 ± 5.6 | 0.03 |
| RBF, μL/min | 10.2 ± 4.7 | 9.0 ± 4.2 | 0.15 |
| $V_{max}$, mm/s | 35.2 ± 9.7 | 32.8 ± 13.2 | 0.19 |
| $V_{min}$, mm/s | 9.6 ± 2.7 | 7.2 ± 2.6 | 0.001 |
| PR | 3.7 ± 0.6 | 4.8 ± 1.7 | 0.001 |
| RI | 0.72 ± 0.04 | 0.77 ± 0.08 | 0.003 |

Abbreviations: DM, diabetes mellitus; PR, pulsatility ratio; RBF, retinal blood flow; RI, resistance index.

*$P$-values comparing control subjects and patients with diabetes.

**Table 3. Characteristics of control subjects and patients with or without insulin treatment.**

|  | Control subjects | Noninsulin group | Insulin group |
|---|---|---|---|
| **Variable** | **(n = 19)** | **(n = 46)** | **(n = 22)** |
| Mean age, years | 57.8 ± 7.9 | 60.2 ± 9.7 | 62.6 ± 8.0 |
| Men/women | 8/11 | 26/20 | 11/11 |
| Glucose, mg/dL | 109.6 ± 27.0 | 142.2 ± 33.6* | 156.4 ± 53.3* |
| HbA1c, % | 5.7 ± 0.3 | 6.8 ± 0.8[†] | 7.0 ± 0.6[†] |
| Duration of diabetes, years |  | 8.3 ± 5.3 | 11.1 ± 7.1 |
| BMI, kg/m$^2$ | 22.1 ± 2.4 | 25.0 ± 5.6* | 25.5 ± 7.3* |
| Systolic BP, mm Hg | 123.1 ± 9.7 | 132.2 ± 15.3 | 129.4 ± 17.8 |
| Diastolic BP, mm Hg | 77.2 ± 7.5 | 77.7 ± 10.6 | 74.2 ± 7.3 |
| Mean BP, mm Hg | 92.5 ± 7.6 | 95.9 ± 10.9 | 92.6 ± 8.1 |
| Heart rate, beats/minute | 71.1 ± 9.7 | 77.4 ± 11.5 | 77.8 ± 11.9 |
| IOP (mm Hg) | 14.7 ± 2.8 | 15.8 ± 2.6 | 14.2 ± 3.0 |
| OPP (mm Hg) | 46.9 ± 4.8 | 48.5 ± 6.5 | 46.8 ± 6.8 |
| Total cholesterol (mg/dL) | 198.5 ± 26.2 | 185.1 ± 25.8 | 195.8 ± 28.0 |
| Triglycerides (mg/dL) | 112.6 ± 47.0 | 129.7 ± 79.8 | 136.5 ± 71.5 |
| HDL (mg/dL) | 62.5 ± 12.1 | 56.6 ± 14.2 | 59.1 ± 14.6 |
| LDL (mg/dL) | 112.3 ± 21.6 | 105.9 ± 24.2 | 110.1 ± 30.3 |
| Blood urea nitrogen (mg/dL) | 12.9 ± 3.1 | 15.0 ± 4.3 | 15.4 ± 3.8 |
| Creatinine (mg/dL) | 0.68 ± 0.14 | 0.73 ± 0.17 | 0.70 ± 0.13 |
| eGFR (mL/min/1.73 m$^2$) | 80.0 ± 11.9 | 77.8 ± 15.0 | 79.1 ± 11.9 |
| Oral antidiabetic drug, no. (%) |  | 38 (83) | 19 (86) |
| Hypertension, no. (%) |  | 18 (39) | 10 (45) |
| Dyslipidemia, no. (%) |  | 27 (59) | 17 (77) |
| Medications, no. (%) |  |  |  |
| Angiotensin II type 1 receptor blocker, no. (%) |  | 15 (33) | 7 (32) |
| Calcium channel antagonist, no. (%) |  | 13 (28) | 8 (36) |
| Statin, no. (%) |  | 27 (59) | 13 (59) |
| Metformin, no. (%) |  | 20 (43) | 13 (59) |
| Dipeptidyl peptidase-4 inhibitor, no. (%) |  | 25 (54) | 7 (32) |

Abbreviations: BMI, body mass index; BP, blood pressure; DM, diabetes mellitus; eGFR, estimated glomerular filtration rate; HbA1c, hemoglobin A1c; HDL, high-density lipoprotein; IOP, intraocular pressure; LDL, low-density lipoprotein; OPP, ocular perfusion pressure.

*Significant ($P < 0.05$) versus the control group (unpaired t-test with Welch's correction).

Contrary to this result of type 2 diabetes, retinal arterial velocity pulsatility was reduced in type 1 diabetes patients with early retinopathy compared to control groups. This difference may be due to type of diabetes.

In the present study, pulse wave analysis revealed that $V_{min}$, but not $V_{max}$, was significantly different between patients with type 2 diabetes and nondiabetic subjects. This significant difference between the two groups coincided with our previous data using laser Doppler velocimetry [4]. Thus, retinal vascular resistance can increase via decreased $V_{min}$ in patients with type 2 diabetes.

The present study using DOCT flowmetry demonstrated no significant difference in retinal blood flow between patients with type 2 diabetes and nondiabetic subjects (Table 1). However, several investigations have demonstrated a disturbance in retinal blood flow in patients with DM [8, 23]. Moreover, our previous investigation in patients with type 2 diabetes with early retinopathy demonstrated abnormalities of retinal blood flow using laser Doppler velocimetry

**Table 4. Retinal circulatory parameters in control subjects and patients with or without insulin treatment.**

| | Control subjects | Noninsulin group | Insulin group |
|---|---|---|---|
| Variable | (n = 19) | (n = 46) | (n = 22) |
| Vessel diameter, μm | 100.1 ± 17.1 | 101.3 ± 18.4 | 96.7 ± 12.3 |
| Blood velocity, mm/s | 19.0 ± 4.9 | 15.6 ± 4.0 | 18.1 ± 8.1 |
| RBF, μL/min | 10.2 ± 4.7 | 8.9 ± 3.8 | 9.1 ± 5.2 |
| $V_{max}$, mm/s | 35.2 ± 9.7 | 30.3 ± 10.0 | 37.9 ± 17.3[†] |
| $V_{min}$, mm/s | 9.6 ± 2.7 | 7.3 ± 2.2* | 7.1 ± 3.2* |
| PR | 3.7 ± 0.6 | 4.3 ± 1.2 | 5.8 ± 2.2*[†] |
| RI | 0.72 ± 0.04 | 0.75 ± 0.07 | 0.80 ± 0.07*[†] |

Abbreviations: PR, pulsatility ratio; RBF, retinal blood flow; RI, resistance index.

*Significant ($P < 0.05$) versus control group (unpaired t-test with Welch's correction).

[†]Significant ($P < 0.05$) versus noninsulin group (unpaired t-test with Welch's correction).

[4]. This inconsistency among previous reports may be attributed to the relatively small sample sizes or differences in measurement techniques. To confirm the exact relationship between these factors, more data are necessary.

PR and RI increased in the insulin group as compared with the noninsulin group, suggesting that long-term insulin therapy can influence the rigidity of the retinal arteries in patients with early DR. Typically, an elevated PR or RI indicates decreased vascular compliance and/or increased vascular resistance, distal to the measurement site, which is close to the capillary bed [24]. If retinal vascular resistance is elevated without a corresponding change in OPP, the retinal vessel parameter must change. However, the vessel diameter, blood velocity, and retinal blood flow were not significantly different between the insulin and noninsulin groups. Our findings of no significant change in the retinal vessel parameters, such as vessel diameter, blood velocity, and retinal blood flow, among these groups indicated that long-term insulin treatment can exert an increased stiffness of the large retinal artery. In addition, this increased stiffness can be ascribed to elevated $V_{max}$ values in the insulin groups. Pulse wave parameter in

**Table 5. Characteristics of 68 patients with type 2 diabetes with early-stage DR and correlations of PR and RI with systemic and ocular parameters.**

| | PR | | RI | |
|---|---|---|---|---|
| Variable | r | P value | r | P-value |
| Age | 0.481 | <0.001 | 0.533 | <0.001 |
| HbA1c | 0.095 | 0.44 | 0.047 | 0.72 |
| Duration of diabetes | 0.381 | 0.002 | 0.364 | 0.003 |
| BMI | 0.046 | 0.71 | 0.028 | 0.83 |
| Systolic BP | −0.147 | 0.23 | −0.127 | 0.30 |
| Diastolic BP | −0.326 | 0.007 | −0.434 | <0.001 |
| Mean BP | −0.291 | 0.02 | −0.354 | 0.003 |
| LDL | 0.026 | 0.83 | 0.031 | 0.80 |
| eGFR | −0.017 | 0.89 | 0.056 | 0.65 |
| Insulin status | 0.432 | <0.001 | 0.341 | 0.004 |
| Vessel diameter, μm | −0.062 | 0.62 | −0.091 | 0.45 |

Abbreviations: BMI, body mass index; BP, blood pressure; DM, diabetes mellitus; eGFR, estimated glomerular filtration rate; HbA1c, hemoglobin A1c; LDL, low-density lipoprotein; PR, pulsatility ratio; RI, resistance index.

**Table 6. Multiple regression analysis of PR and RI in 68 patients with type 2 DM.**

| Variable | PR | RI |
|---|---|---|
| Age | 0.377 (<0.001) | 0.384 (<0.001) |
| Insulin use | 0.322 (0.002) | 0.196 (0.06) |
| Duration | 0.218 (0.03) | 0.204 (0.04) |
| Diastolic BP | −0.028 (0.80) | −0.169 (0.13) |
| | $r^2 = 0.413$ $P = 0.0001$ | $r^2 = 0.419$ $P = 0.0001$ |

Abbreviations: DM, diabetes mellitus; BP, blood pressure; PR, pulsatility ratio; RI, resistance index.

the microvessels can be at least two to three orders of magnitude slower than that in the macrovessels [25]. However, owing to the linear pulse wave transmission theory on a branch system of vessels [19, 26], RI and PR in the current study appear to be consistent with the contemporary vascular physiology.

Most research has confirmed the vasodilator activity of insulin in porcine retinal arterioles [27]. However, these experiments have been performed using a nondiabetic animal model. Because the insulin receptor of tissue in the insulin-resistant condition was downregulated [28], vasoreactivity by insulin may depend on the clinical state of hyperinsulinemia, which is typically noted in patients with type 2 diabetes mellitus. In general, an infusion of insulin decreased the augmentation index, a measure of arterial stiffness, in nonobese subjects, whereas this insulin-reduction effect of stiffness of the large artery was diminished in obese subjects, which is characterized by insulin resistance status [29]. An experimental investigation also demonstrated that the use of insulin stimulated the proliferation of vascular smooth muscle cells, which is an important step in the pathogenesis of atherosclerotic plaque [2], from aortas of insulin-resistant spontaneously hypertensive rats [6]. Thus, although we did not measure the serum insulin concentrations or insulin RI in the present study, insulin-induced retinal vasoreactivity can decrease under insulin-resistant states, such as type 2 diabetes.

Insulin is susceptible to glycation by glucose, D-ribose, and other highly reactive carbonyls, such as methylglyoxal, especially under diabetic conditions [30]. Recently, an experimental study revealed that glycated insulin by D-ribose induced the formation of advanced glycation end products in endothelial cells [31]. Moreover, our investigation using laser Doppler velocimetry revealed that PR was positively related to the concentration of serum advanced glycation end products in patients with nonproliferative DR [22]. Therefore, although we did not examine the plasma concentrations of advanced glycation end products in the present study, these findings indicate that the mechanism of arterial stiffness in patients with type 2 diabetes may be the accumulation of advanced glycation end products due to long-term insulin treatment.

Age, which is a powerful predictor of arteriosclerosis [32], can affect the retinal vascular parameter obtained from the velocity profile, as previously described in our study using laser Doppler velocimetry [33]. Moreover, a previous study reported that older volunteers had a higher retinal pulse wave velocity, which is an indicator of retinal arterial stiffness, than younger volunteers did [21]. We confirmed here that this effect of aging on retinal arterial stiffness is also the case for patients with type 2 diabetes with early retinopathy. Thus, we speculated that aging can aggravate the rigidity of the retinal arterioles in type 2 diabetic patients with early retinopathy.

In the present study, multivariate analysis revealed a positive association between PR, RI, and diabetes duration in patients with type 2 diabetes, suggesting that diabetes duration can alter the stiffness of the retinal arterioles. A previous report demonstrated that the diabetes

duration was associated with increased aortic stiffness in patients with type 2 diabetes [34]. Thus, diabetes duration can affect arterial properties in not only macrovascular but also microvascular lesions. In addition, diabetes duration was strongly correlated with the development or progression of DR in patients with type 2 diabetes [35]. Therefore, we speculated that a longer duration of diabetes may influence the development or progression of DR via the increased arterial stiffness in the retinal arterioles.

In the diabetic retina, immunoreactivity for N [ε]-(carboxylmethyl) lysine, an advanced glycation end product, was observed in the vascular wall, whereas immunohistochemistry for smooth muscle actin, a marker of smooth muscle cells, was attenuated [36]. Moreover, smooth muscle cells, the main constituents of the vascular wall, are damaged and replaced by collagen bundles in the large vessels of retinas in advanced diabetes. Thus, increased PR and RI can reflect altered smooth muscle cells in the retinal arterioles of patients with diabetes.

Some evidence indicates that insulin treatment in patients with diabetes can affect the coagulatory system [37, 38]. Moreover, an experimental report demonstrated that insulin increased the plasminogen activator inhibitor 1(PAI-1) concentration, which can be an important component of the coagulatory system, in endothelial cells cocultured with smooth muscle cells from the human arterial segment [39]. The present study reported an increased PR and RI without vessel parameter changes, such as vessel diameter, blood velocity, and RBF, in patients with type 2 diabetes with long-term insulin treatment. Thus, these findings indicated that blood properties, such as coagulation, but not vessel parameters, can be altered in the retinal artery with insulin-treatment induced elevated stiffness. Therefore, although the serum concentration of the fibrinolytic system components, including PAI-1, was not examined, impaired fibrinolysis resulting from long-term insulin treatment in the retinal artery may be implicated in the development and progression of DR.

The present study has several limitations. First, the cross-sectional study design limits the causal inference related to the associations observed. Further prospective studies are required to investigate the relationship between long-term insulin treatment and retinal arterial stiffness. Second, the present study could not assess medications in detail and, therefore, might have overlooked the possible pleiotropic effects of certain drugs, such as antihypertensive and lipid-modifying drugs. In particular, the potential confounding effect caused by statins, which may be potent retinal vasodilators, should be estimated. The association of PR with RI without using statins between DM subgroups had a similar trend, albeit not significant (PR, $P = 0.09$; RI, $P = 0.11$ insulin group versus noninsulin group by unpaired t-test with Welch's correction, data not shown). Third, patients with severe hyperglycemia required long-term external insulin treatment. Although we found no difference in plasma glucose and HbA1c between the two groups (insulin and noninsulin groups), increased arterial stiffness in the retinal arterioles in the insulin group may reflect the vascular damage induced by hyperglycemia. Fourth, it remains unclear whether there is a linear dose effect of insulin on retinal vascular stiffness. In the present study, no relationship between PR or RI and duration of insulin use was found (Pearson coefficient of correlation, PR: $r = 0.252$, $P = 0.26$; RI: $r = 0.335$, $P = 0.13$, data not shown). However, because of the variations between patients in the amount of daily insulin used, number of daily insulin doses, and type of insulin, such as short acting or long acting, it is difficult to examine the dose-dependent effect of insulin on the retinal vessel parameters in this clinical study. Fifth, flow velocity can influence the vessel diameter measurement. The OCT device enables the measurement of vessel diameter by detecting the red blood cells flowing at the vessel border. However, it is possible that the OCT flowmeter used in this study could not detect very-low-velocity flow of red blood cells at the vessel border, owing to phase noise velocity profiles. Although our in vitro study using glass capillaries with an inner diameter of 140 μm suggested that the accuracy of the measured diameter is within a couple of

microns [10], the difficulty in detection of low velocity flow can affect the measurement of human retinal vessel diameters.

## Conclusion

We found that PR and RI were associated with increased vascular rigidity of the retinal artery in patients with early retinopathy who were taking insulin as compared with those not taking insulin. Because a previous study reported a significant association between insulin treatment and DR, these findings could indicate an involvement of increased PR and RI associated with retinal arterial stiffness, owing to long-term insulin treatment in the pathogenesis of DR.

## Author Contributions

**Conceptualization:** Tsuneaki Omae, Youngseok Song, Takafumi Yoshioka, Akitoshi Yoshida.

**Data curation:** Tsuneaki Omae, Takafumi Yoshioka, Tomofumi Tani.

**Formal analysis:** Tsuneaki Omae, Youngseok Song.

**Funding acquisition:** Tsuneaki Omae.

**Investigation:** Tsuneaki Omae.

**Methodology:** Tsuneaki Omae, Tomofumi Tani.

**Project administration:** Akitoshi Yoshida.

**Resources:** Tsuneaki Omae.

**Software:** Tsuneaki Omae.

**Supervision:** Youngseok Song, Tomofumi Tani, Akitoshi Yoshida.

**Validation:** Tsuneaki Omae, Youngseok Song, Takafumi Yoshioka.

**Writing – original draft:** Tsuneaki Omae.

**Writing – review & editing:** Tsuneaki Omae.

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
