## [Decision Letter · Decision Letter 0]

11 Jan 2021

PONE-D-20-38250

Effect of insulin treatment on retinal arterial stiffness in patients with type 2 diabetes

PLOS ONE

Dear Dr. Omae,

Thank you for submitting your manuscript to PLOS ONE. After careful consideration, we feel that it has merit but does not fully meet PLOS ONE’s publication criteria as it currently stands. Therefore, we invite you to submit a revised version of the manuscript that addresses the points raised during the review process.

The method you used is suitable for measuring velocities in large retinal arteries but it has not been validated to measure retinal arterial stiffness itself. Please revise your statistical analysis as recommended by the reviewers.

We look forward to receiving your revised manuscript.

Kind regards,

Alfred S Lewin, Ph.D.

Academic Editor

PLOS ONE

Journal Requirements:

2.Thank you for including your ethics statement: 

"all information entered here is included in the Methods section of the manuscript".   

4.Thank you for stating the following in the Funding Section of your manuscript:

"This study was supported by a Grant-in-Aid for Scientific Research (C) 16K11311 (TO) and 20K09764 (TO) from the Ministry of Education, Science, and Culture, Tokyo, Japan."

 "The author(s) received no specific funding for this work"

5.We noticed you have some minor occurrence of overlapping text with the following previous publications, which needs to be addressed:

- http://amcor.asahikawa-med.ac.jp/modules/xoonips/download.php/27002294.pdf?file_id=8400

- https://iovs.arvojournals.org/article.aspx?articleid=2657329

In your revision ensure you cite all your sources (including your own works), and quote or rephrase any duplicated text outside the methods section. Further consideration is dependent on these concerns being addressed.

Reviewers' comments:

Reviewer's Responses to Questions

**Comments to the Author**

1. Is the manuscript technically sound, and do the data support the conclusions?

Reviewer #1: Partly

Reviewer #2: Yes

2. Has the statistical analysis been performed appropriately and rigorously? 

Reviewer #1: No

Reviewer #2: N/A

3. Have the authors made all data underlying the findings in their manuscript fully available?

Reviewer #1: Yes

Reviewer #2: Yes

4. Is the manuscript presented in an intelligible fashion and written in standard English?

Reviewer #1: Yes

Reviewer #2: Yes

5. Review Comments to the Author

Reviewer #1: The authors are evaluating whether a long-term insulin treatment is associated with abnormalities in retinal circulation in type 2 diabetic patients under the insulin therapy and without the therapy. Quantitative parameters of the measured velocity pulse wave during a cardiac cycle show significant alteration in the retinal arterial circulation in patients with diabetes mellitus type 2. Although the results of the study seem very interesting for clinical applications and retinal blood flow research, the measurement technique does not allow to report directly on the retinal arterial stiffness. Moreover the data analysis and its interpretation flaw.

Major remarks

The title of the paper claims to show effect of insulin treatment on retinal arterial stiffness. However, this supposition of the authors is not directly related to the measurement technique used. The cited references show the method being able to assess velocities in large retinal arteries but it has not ever been validated to measure retinal arterial stiffness itself! The claim of the authors to assess the retinal arterial stiffness on the base of the formulas from the macrocirculation is reasonable but still far-fetched. This reviewer suggests to be honest and to report objectively what was really measured: blood velocities in the retinal artery during the cardiac cycle and related compliance parameters. It is reasonable to mention correctly and carefully in the discussion and probably in the conclusion, that some of these parameters might relate to arterial stiffness/rigidity. However, is too ambitious to postulate this directly in the title. It should be changed and the corresponding ambitious affirmations in the abstract and in the conclusion should be adjusted as well.

The discussion section would benefit from more considerations on the etiology and structural changes of the velocity wave form in the retinal arteries and their relation to the stiffness (usually they work with pressure wave form in the PWA). What physiological/ pathophysiological meaning do the waveform changes have? Why the reported parameters and not others were chosen to describe/represent the pulse wave? What do the authors mean with the alteration in the stiffness: a structural remodeling of the arterial wall? A functional constriction of retinal arterioles downstream to the measured arterial segment?

It is advisable to check the correlation of velocity parameters to the vessel diameter. This important interaction is forgotten in Table 5. Even if the vessel diameters were not significantly different in the groups, the insulin group showed smaller diameters and dependent to the flow characteristics the diameter comes in the calculation of the velocity to the power of ~2.

The medical ethics issue is questionable: The authors report: “This study complied with the guidelines approved by the ethics Committee of …” Was the study approved or only “complied with”? What institution?

Measuring issues:

How accurate the vessel diameter was measured? Was the technique able to measure in µm without correction to the AL that is not done by conventional OCT-devices? The discrepancy to the real value in µm can thus vary in +/- 10%! Additionally the measurement accuracy of 0.1 µm (as reported with the numbers) is incredible for the modern OCT-technology! How was the real precision of the vessel diameter measurement? The issue is very important and can partially explain the unusual controversial result of similar blood flow rates in diabetic and non-diabetic subjects in the study.

Statistical issues:

The sample sizes of the examined groups are considerably different. The use of t-test is known to be sensible for unequal variances especially for very different sample sizes. Welch’s test should be used instead. The use of ANOVA implies the homoscedasticity as well. As it can be seen from the data, for the key “vascular resistance” parameters in the study the variances are quite different! E.g. the variance of PR is 0.36 in the control group and 4.6 in the insulin group: more than 10 times! (Table 4). Moreover, there are 7 circulatory parameters reported in Tab. 2 with their p-values. To report a solid result on statistical significance one should correct to multiple comparisons here as well (e.g. using Bonferroni correction). The authors are encouraged to apply correct statistical testing, that probably reconsiders their results.

Line 282: “Typically, an elevated PR or RI indicates decreased vascular compliance and/or increased vascular resistance, distal to the measurement site.[21]”

The affirmation is very general and somehow inaccurate regarding the cited literature, since the pressure-volume relationship, vessel structure and related aspects are different in the microcirculation compared to the macrocirculation. What is meant exactly with “distal to retinal arterioles” that are themselves quite close to the capillary bed?

Minor remarks:

Line 178: “Pulse wave analysis revealed an increase in the Vmin, but not Vmax, in the diabetic group and a greater PR and RI in diabetic patients.” A confusing affirmation with the view to the results in table 2: Vmin is smaller there in DM group

English of the manuscript should be improved. Some formulations in the manuscript are overall inaccurate and therefore sometimes not understandable, e.g.:

Line 45 and further on: “Retinal vessel rigidity”. Consider „arterial“. The venous rigidity was not measured at all!

Line 62: “optimal imaging technology”. What do the authors mean?

Line 85: “Patients with blood pressure (BP) level…were considered as hypertension.” Or further on: “Patients…. as dyslipidemia.”

Patients cannot be a disease!

Line 92: “Patients with uncontrolled controlled diabetes.”

What does it mean???

Line 95: “a history renal dysfunction or cardiovascular…”

Consider history of!

Line 285: “However, the retinal blood parameter was not significantly different between the insulin and noninsulin groups.”

What retinal blood parameters? Especial parameters of retinal blood?

Line 96 and others: Articles are frequently missed!

Line 258: “Doppler velocimetry also indicated that PR was significantly greater, without significant differences in vessel parameters.”

Quite inaccurate formulation. What are “vessel parameters”? Does the velocity belong to it?

Line 286: “Our findings of no significant change in the retinal vessel parameter across these groups…”

In what parameter exactly?

Reviewer #2: This article evaluated whether long-term insulin treatment is associated with abnormalities in retinal circulation in type 2 diabetic patients. Overall, the article is well written with interesting findings.

Here are my concerns:

1. Methods section, subsection subjects - How was IOP measured? Applanation tonometry?

2. Methods section, subsection subjects -The insulin group was defined as those patients receiving insulin therapy for at least 1 year. Although authors mention long term insulin treatment, the duration of insulin treatment was not mentioned. Were they using insulin treatment throughout disease duration?

3. Statistical analysis: The group on insulin treatment very small. Did authors do a formal sample size calculation?

4. Results section: Were confounding factors taken into consideration when evaluating the relationship between PR and insulin use? If yes, please include.

5. Conclusion section: How will increase in retinal arterial stiffness (owing to long-term insulin treatment) without affecting the diameter, velocity, or blood flow be implicated in the pathogenesis of DR. Please elaborate.

6. Typo error: Line 94, Delete ‘controlled’ between ‘uncontrolled’ and ‘diabetes’

7. Table 1 and 2 not very useful since the information on control group is duplicated in Table 3 and 4.

6. PLOS authors have the option to publish the peer review history of their article (what does this mean?). If published, this will include your full peer review and any attached files.

Reviewer #1: No

Reviewer #2: No

---

## [Author Response · Author response to Decision Letter 0]

7 Jun 2021

Reviewer 1 : I have incorporated all of your suggestions into my revision. They were very helpful. Thank you.

Reviewer 2 : I have incorporated all of your suggestions into my revision. Thank you for your help.

---

## [Decision Letter · Decision Letter 1]

8 Jul 2021

Effect of insulin treatment on pulsatility ratio and resistance index of the retinal artery in patients with type 2 diabetes

PONE-D-20-38250R1

Dear Dr. Omae,

We’re pleased to inform you that your manuscript has been judged scientifically suitable for publication and will be formally accepted for publication once it meets all outstanding technical requirements.

Kind regards,

Alfred S Lewin, Ph.D.

Section Editor

PLOS ONE

Additional Editor Comments (optional):

Reviewers' comments:

Reviewer's Responses to Questions

**Comments to the Author**

1. If the authors have adequately addressed your comments raised in a previous round of review and you feel that this manuscript is now acceptable for publication, you may indicate that here to bypass the “Comments to the Author” section, enter your conflict of interest statement in the “Confidential to Editor” section, and submit your "Accept" recommendation.

Reviewer #2: All comments have been addressed

2. Is the manuscript technically sound, and do the data support the conclusions?

Reviewer #2: Yes

3. Has the statistical analysis been performed appropriately and rigorously? 

Reviewer #2: I Don't Know

4. Have the authors made all data underlying the findings in their manuscript fully available?

Reviewer #2: Yes

5. Is the manuscript presented in an intelligible fashion and written in standard English?

Reviewer #2: Yes

6. Review Comments to the Author

Reviewer #2: The revised version of the manuscript is much better. All my queries have been addressed in a satisfactorily manner.

7. PLOS authors have the option to publish the peer review history of their article (what does this mean?). If published, this will include your full peer review and any attached files.

Reviewer #2: No

---

## [Editor Report · Acceptance letter]

12 Jul 2021

PONE-D-20-38250R1 

Effect of insulin treatment on pulsatility ratio and resistance index of the retinal artery in patients with type 2 diabetes 

Dear Dr. Omae:

I'm pleased to inform you that your manuscript has been deemed suitable for publication in PLOS ONE. Congratulations! Your manuscript is now with our production department. 

Kind regards, 

on behalf of

Dr. Alfred S Lewin 

Section Editor

PLOS ONE